# A unifying structural and functional model of the coronavirus replication organelle: Tracking down RNA synthesis

Eric J. Snijder[1]*, Ronald W. A. L. Limpens[2], Adriaan H. de Wilde[1]¤, Anja W. M. de Jong[2], Jessika C. Zevenhoven-Dobbe[1], Helena J. Maier[3], Frank F. G. A. Faas[2], Abraham J. Koster[2], Montserrat Bárcena[2]*

**1** Molecular Virology Laboratory, Department of Medical Microbiology, Leiden University Medical Center, Leiden, the Netherlands, **2** Section Electron Microscopy, Department of Cell and Chemical Biology, Leiden University Medical Center, Leiden, the Netherlands, **3** The Pirbright Institute, Pirbright, Surrey, United Kingdom

¤ Current address: Janssen Vaccines and Prevention, Pharmaceutical Companies of Johnson and Johnson, Leiden, the Netherlands
* m.barcena@lumc.nl (MB); e.j.snijder@lumc.nl (EJS)

## Abstract

Zoonotic coronavirus (CoV) infections, such as those responsible for the current severe acute respiratory syndrome-CoV 2 (SARS-CoV-2) pandemic, cause grave international public health concern. In infected cells, the CoV RNA-synthesizing machinery associates with modified endoplasmic reticulum membranes that are transformed into the viral replication organelle (RO). Although double-membrane vesicles (DMVs) appear to be a pan-CoV RO element, studies to date describe an assortment of additional CoV-induced membrane structures. Despite much speculation, it remains unclear which RO element(s) accommodate viral RNA synthesis. Here we provide detailed 2D and 3D analyses of CoV ROs and show that diverse CoVs essentially induce the same membrane modifications, including the small open double-membrane spherules (DMSs) previously thought to be restricted to gamma- and delta-CoV infections and proposed as sites of replication. Metabolic labeling of newly synthesized viral RNA followed by quantitative electron microscopy (EM) autoradiography revealed abundant viral RNA synthesis associated with DMVs in cells infected with the beta-CoVs Middle East respiratory syndrome-CoV (MERS-CoV) and SARS-CoV and the gamma-CoV infectious bronchitis virus. RNA synthesis could not be linked to DMSs or any other cellular or virus-induced structure. Our results provide a unifying model of the CoV RO and clearly establish DMVs as the central hub for viral RNA synthesis and a potential drug target in CoV infection.

## Introduction

The RNA synthesis of all positive-stranded RNA (+RNA) viruses of eukaryotes occurs in the cytoplasm of the host cell, in conjunction with modified endomembranes that are often

**Data Availability Statement:** All relevant data are within the paper and its Supporting Information files.

**Funding:** This research was partially funded by the Netherlands Organization for Scientific Research (NWO, https://www.nwo.nl/en) through grants to M.B. (NWO-MEERVOUD-863.10.003) and to E.J.S. and A.J.K. (NWO-CW-700.57.301). The funder had no role in study design, data collection and analysis, decision to publish, or preparation of the manuscript.

**Competing interests:** The authors have declared that no competing interests exist.

**Abbreviations:** +RNA, positive-stranded RNA; CM, convoluted membranes; CoV, coronavirus; DMS, double-membrane spherule; DMV, double-membrane vesicle; ds, double-stranded; EM, electron microscopy; ER, endoplasmic reticulum; ERGIC, ER-Golgi intermediate compartment; HCoV-229E, human coronavirus 229E; HCV, hepatitis C virus; HPF-FS, high-pressure freezing and freeze-substitution; hpi, hours postinfection; IBV, infectious bronchitis virus; IEM, immunoelectron microscopy; MERS-CoV, Middle East respiratory syndrome-CoV; MHV, murine hepatitis virus; MOI, multiplicity of infection; nsp, nonstructural protein; RLI, relative labeling index; RO, replication organelle; SARS-CoV, severe acute respiratory syndrome-CoV; vRNA, viral RNA.

referred to as viral replication organelles (ROs) [1–3]. ROs are generally believed to provide tailored platforms that facilitate viral replication by concentrating relevant factors and spatially organizing distinct steps in the viral cycle. Additionally, ROs may contribute to the evasion of cellular innate immune defenses that detect viral RNA (vRNA) [4].

Two main RO prototypes have been discriminated: small spherular invaginations and large (r) vesiculotubular clusters consisting of single- and/or double-membrane structures, to which viral replicative proteins and specific host factors can be recruited. The formation of invaginations can occur at the membrane of various organelles, including endoplasmic reticulum (ER), endolysosomes, and mitochondria [5–9]. The lumen of the resulting microcompartment is connected with the cytosol by a "neck-like" channel that can mediate transport of metabolites and export of newly made positive-sense vRNAs to the cytosol for translation and packaging. In general, the morphological and functional characterization of ROs of the second, vesiculo-tubular type is lagging behind. Such structures, which always include double-membrane vesicles (DMVs), commonly derive from membranes of the secretory pathway and have been found in cells infected with, e.g., picornaviruses [10–11], noroviruses [12], hepatitis C virus (HCV) [13], and different nidoviruses, including the arterivirus and coronavirus (CoV) families [14–18].

The first electron tomography analysis of a CoV-induced RO, that of the severe acute respiratory syndrome-CoV [15] (SARS-CoV, genus *Betacoronavirus*), raised a variety of functional considerations. Intriguingly, while double-stranded (ds) RNA, a presumed intermediate and marker for vRNA synthesis [19], was found inside the virus-induced DMVs, these lacked visible connections to the cytosol [15]. vRNA synthesis inside fully closed DMVs would pose the conundrum of how metabolites and newly made genomic and subgenomic mRNAs could be transported across the double-lipid bilayer. Importantly, dsRNA is not a bona fide marker for vRNA synthesis because it may no longer be associated with the active enzymatic replication complexes in which most of the 16 viral nonstructural proteins (nsps) come together. Thus, the possibility of vRNA synthesis taking place in alternative locations, such as the convoluted membranes (CM) that are also prominent elements of the beta-CoV RO [15,20–21], was entirely possible and started to attract attention. Notably, DMVs can be also formed in the absence of vRNA synthesis by expression of key transmembrane nsps [22–23]. Moreover, several studies suggested a lack of direct correlation between the number of DMVs and the levels of CoV replication in the infected cell [24–25].

The interpretation of the CoV RO structure and function was further compounded by the discovery of different RO elements (never reported in beta-CoV infections) that set apart other distantly related CoVs genera. In particular, zippered ER (instead of CM) and double-membrane spherules (DMSs) were detected for the avian gamma-CoV infectious bronchitis virus (IBV) [26] and, recently, for the porcine deltacoronavirus [27]. The size and topology of these DMSs, which were invaginations in the zippered ER, were remarkably similar to those of the spherular invaginations induced by other +RNA viruses, and consequently, DMSs were suggested to be sites of vRNA synthesis [26].

In this work, we provide an in-depth analysis of the structure and function of the RO induced by members of different CoV genera, with a special focus on beta-CoVs such as SARS-CoV and Middle East respiratory syndrome-CoV (MERS-CoV) [28–29]. Although the SARS-CoV outbreak was contained in 2003, MERS-CoV has continued to pose a serious zoonotic threat to human health since 2012. A previously unknown beta-CoV (SARS-CoV-2), which emerged in China at the end of 2019 [30], is responsible for the current pandemic that is shaking societies and economies. The SARS-CoV-2 genome sequence is 79.5% identical to that of SARS-CoV [31], yielding 86% overall nsp sequence identity and suggesting strong functional similarities in the replication of both viruses.

Our observations on the 3D morphology of the MERS-CoV RO were compared with data from cells infected with other alpha-, beta-, and gamma-CoVs. This comparative analysis made it clear that all these CoV induce essentially the same membrane structures, including DMSs. Metabolic labeling of newly synthesized vRNA was used to determine the site(s) of vRNA synthesis within the CoV RO. To this end, we used a radiolabeled nucleoside ([³H]uridine) and applied the classic and highly sensitive technique of EM autoradiography [32–33] in combination with advanced quantitative analysis tools. This approach revealed that DMVs are the primary site of CoV RNA synthesis, with neither DMSs nor CM nor zippered ER being labeled to a significant extent. Our study provides a comprehensive and unifying model of the CoV RO structure. It also returns DMVs to center stage as the hub of CoV RNA synthesis and a potential antiviral drug target.

## Results

### MERS-CoV induces a membrane network of modified membranes that contains DMSs

We first set out to analyze the ultrastructure of MERS-CoV-infected Huh7 cells under sample preparation conditions favorable for autoradiography (see Materials and methods). To this end, Huh7 cells were infected with MERS-CoV at a multiplicity of infection (MOI) of 5, chemically fixed at 12 hours postinfection (hpi), and further processed for 2D EM and electron tomography (Fig 1, S1 Video). This time point, which represents the late exponential phase of viral replication [21], provided a variety of abundant virus-induced membrane structures. Strikingly, in addition to the DMVs and CM that are well-established hallmarks of beta-CoV infections, the presence of small spherules, occasionally in large numbers, was readily apparent (Fig 1A and 1B). These spherules were notably similar to the DMSs previously described for the gamma-CoV IBV [26]. Their remarkably regular size of approximately 80 nm (average diameter 79.8 ± 2.5 nm, $n = 58$), a delimiting double membrane, and their electron-dense content made these spherules clearly distinct from other structures, including progeny virions, which had comparable diameter (Fig 1C and 1D, S1 Data).

The DMSs generated during IBV infection were previously described as invaginations of the zippered ER that remain open to the cytosol [26]. In MERS-CoV-infected cells, the DMSs were connected to the CM from which they seemed to derive (Fig 1E). Clear openings to the cytosol could not be detected for the large majority (around 80%, $n = 54$) of the fully reconstructed DMSs, which suggests that the original invagination may eventually transform into a sealed compartment. This type of apparently closed DMSs were also present, though in a lower proportion (around 50%, $n = 39$), in IBV-infected cell samples processed in an identical manner (Fig 2).

Our data suggest a functional analogy between zippered ER and CM, as both structures appear to provide membranes for DMS formation. In fact, both bore a striking resemblance under the same sample preparation conditions: MERS-CoV-induced CM noticeably consisted of zippered smooth membranes that were connected to the rough ER and branched and curved in intricate arrangements. Although unbranched zippered ER was most common in IBV-infected cells (Fig 2A), as described [26], we also detected zippered ER morphologically closer to CM (Fig 2B). These observations argue for zippered ER and CM representing alternative configurations of essentially the same type of virus-induced modification.

The 3D architecture of MERS-CoV-induced RO aligned with previous observations for other CoVs [15,26]. In our samples, the 2 delimiting membranes of the DMVs had a distended appearance, in contrast with the tight membrane apposition observed in samples prepared by high-pressure freezing, freeze-substitution (HPF-FS) [21]. Although constrictions in this

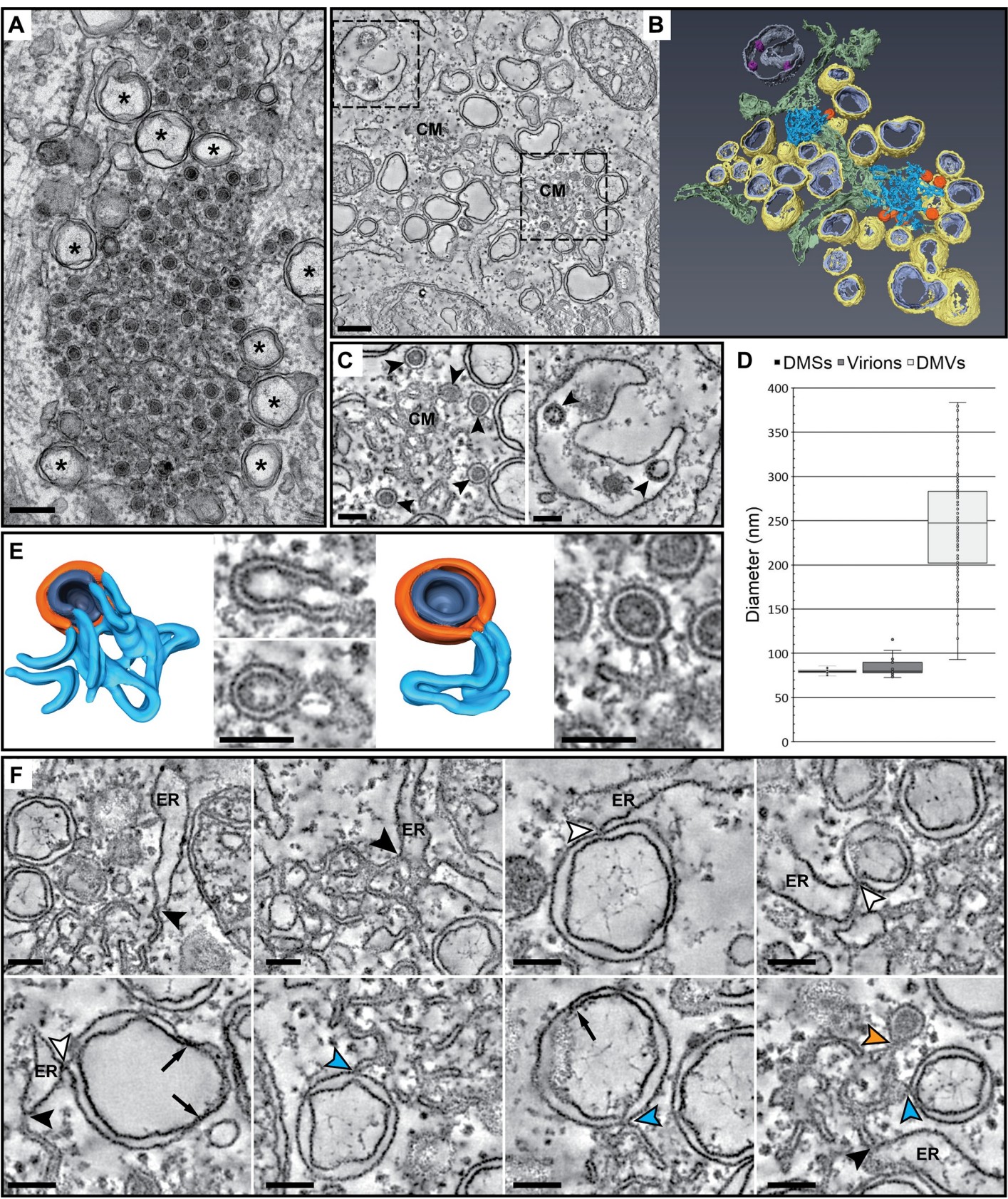

**Fig 1. Membrane structures induced by MERS-CoV infection.** Electron microscopy analysis of Huh7 cells infected with MERS-CoV (MOI 5, 12 hpi). (A) Electron micrograph of an area with abundant DMSs. DMVs (asterisks) are interspersed and surrounding the DMS cluster. (B) Slice through a tomogram (left) and corresponding surface-rendered model (right) of a representative area containing the different types of MERS-CoV-induced membrane modifications: CM (blue), DMSs (orange), and DMVs (yellow and lilac, outer and inner membranes, respectively). The model also highlights ER membranes (green) and a vesicle (silver) containing new virions (pink). (See also S1 Video.) (C) Comparison of DMSs and virions (arrowheads in left and right panels, respectively) in enlarged views of tomographic slices from the regions boxed in (B). The DMSs are similar in size but distinct in appearance from newly formed MERS-CoV particles. (D) Whisker plots of the size distribution of DMSs ($n$ = 58), virions ($n$ = 28), and DMVs ($n$ = 109), as measured from the tomograms. DMSs and virions have a comparable size (median diameter, 80 nm), whereas the median diameter of the DMVs is 247 nm (S1 Data). (E) Models and tomographic slices through an open (left) and closed (right) DMS. Both types of DMSs are connected with the CM. In open DMSs, both the inner and outer membranes (dark blue and orange, respectively) are continuous with CM. Two slices approximately 8 nm apart in the reconstruction are shown. For closed DMSs, only the outer membrane is connected to CM, whereas the inner membrane seems to define a closed compartment. (F) Gallery of tomographic slices highlighting membrane connections between different elements of the MERS-CoV RO and of these with the ER. These include CM-ER (black arrowheads), DMV-ER (white arrowheads), CM-DMV (blue arrowheads), and CM-DMS (orange arrowhead) connections. Constrictions in the DMVs are indicated by arrows. Scale bars, 250 nm (A, B), and 100 nm (C-F). CM, convoluted membranes; DMS, double-membrane spherule; DMV, double-membrane vesicle; ER, endoplasmic reticulum; hpi, hours postinfection; MERS-CoV, Middle East respiratory syndrome-coronavirus; MOI, multiplicity of infection; RO, replication organelle.

distended pattern could be frequently observed (Fig 1F, arrows), no clear openings to the cytosol were detected. All 3 types of MERS-CoV-induced membrane modifications appeared to be interconnected, either directly or indirectly through the ER. While DMSs were connected to CM, and CM to ER, ER membranes were often continuous with the outer membrane of the DMVs (Fig 1F, arrowheads). Therefore, like other CoVs, MERS-CoV infection appears to induce a network of largely interconnected modified ER membranes that, as a whole, can be considered the CoV RO.

## Diverse CoVs across different genera induce the same RO elements

Intriguingly, DMSs had never been reported for beta-CoV infections in previous characterizations (including ours) that used different sample preparation conditions and/or different cell lines. This prompted us to revisit those samples for a closer examination. A targeted search for DMSs in MERS-CoV-infected Vero cells [21] and SARS-CoV-infected Vero E6 cells [15]

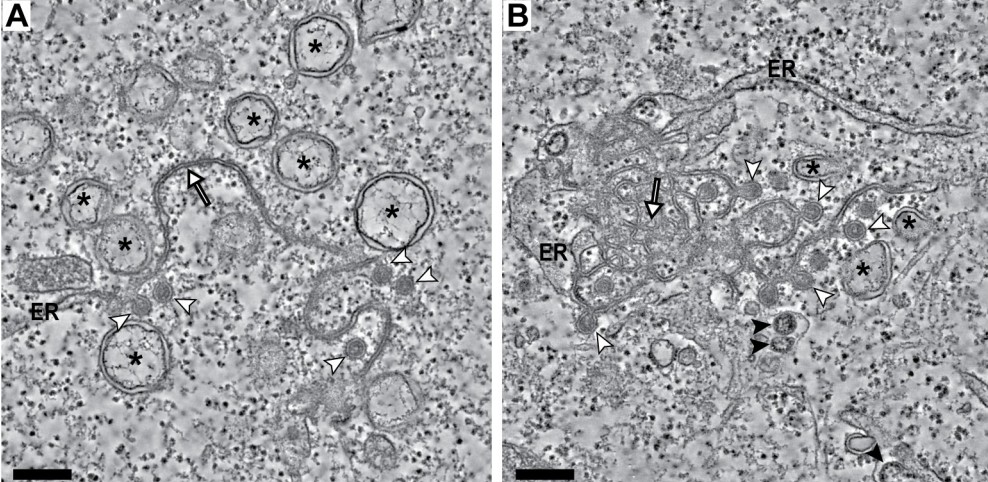

**Fig 2. Membrane structures induced by gamma-CoV infections.** Tomography of Vero cells infected with IBV, fixed at 16 hpi, and processed for EM following the same protocol as for MERS-CoV-infected cells (Fig 1). Tomographic slices through 2 regions containing IBV-induced membrane modifications. These include DMVs (asterisks), DMSs (white arrowheads), and zippered ER (white arrows). Most zippered ER consists of long stretches of ER-derived paired membranes (A), though branching zippered ER, closer to the CM described for beta-CoV, was also present. (B) Virus particles (black arrowheads) budding into the ER membranes were often observed. Scale bars, 250 nm. CM, convoluted membranes; CoV, coronavirus; DMS, double-membrane spherule; DMV, double-membrane vesicle; EM, electron microscopy; ER, endoplasmic reticulum; hpi, hours postinfection; IBV, infectious bronchitis virus; MERS-CoV, Middle East respiratory syndrome-CoV.

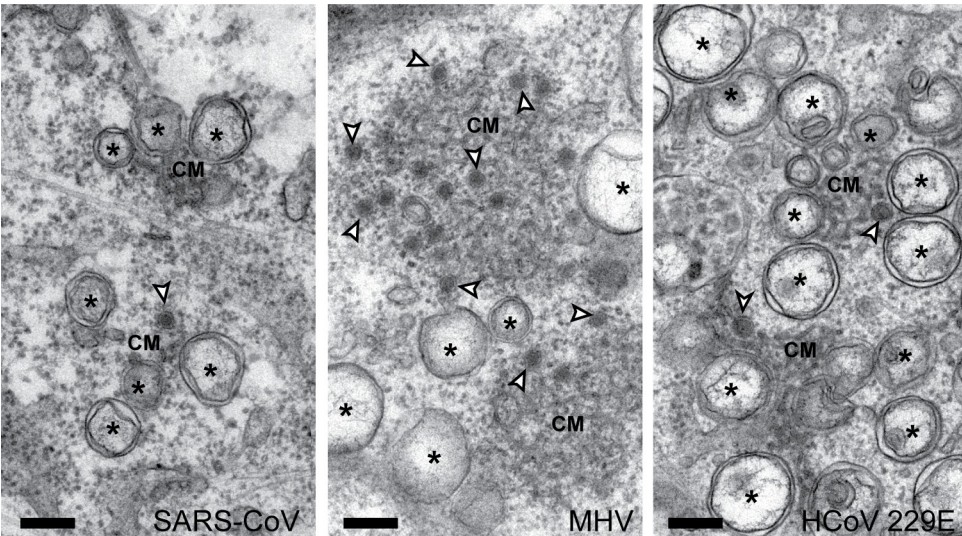

**Fig 3. DMSs are induced by diverse beta- and alpha-CoVs.** 2D-EM images from 100-nm-thick sections of different mammal cells infected with (from left to right) SARS-CoV (MOI 10, 9 hpi), MHV (MOI 10, 8 hpi), and HCoV-229E (MOI 5, 24 hpi). These time points represent intermediate to late stages in infection [34–36]. Both beta-CoVs (A,B) and the alpha-CoV (C) induce membrane modifications that include not only DMVs (asterisks) and CM but also DMSs (white arrowheads). Scale bars, 250 nm. CM, convoluted membranes; CoV, coronavirus; DMS, double-membrane spherule; DMV, double-membrane vesicle; EM, electron microscopy; ER, endoplasmic reticulum; HCoV-229E, human coronavirus 229E; hpi, hours postinfection; IBV, infectious bronchitis virus; MHV, murine hepatitis virus; MOI, multiplicity of infection; SARS-CoV, severe acute respiratory syndrome-CoV.

readily revealed similar DMSs, embedded and somewhat concealed in CM with a denser and more tangled appearance that can be attributed to sample preparation differences (S1 Fig, compare to Figs 1 and 3). To further explore whether these observations could be extended to other CoVs, we analyzed a third beta-CoV (murine hepatitis virus [MHV]) as well as a member of the genus *Alphacoronavirus* (human coronavirus 229E [HCoV-229E]). Both in MHV-infected 17Cl1 cells and in HCoV-229E-infected Huh7 cells, virus-induced DMSs could be detected, with their characteristic size, appearance, and spatial association with CM (Fig 3). These results demonstrate that virus-induced DMSs are not exclusive to some CoV genera or specific cell lines but are instead a common characteristic of CoV-infected cells.

## CoV RNA synthesis is confined to RO regions

To investigate the subcellular localization of vRNA synthesis in cells infected with different CoVs, we metabolically labeled newly synthesized vRNA by arresting cellular transcription with actinomycin D and using a radiolabeled nucleoside precursor ([5-$^3$H]uridine) for subsequent detection by EM autoradiography [32–33] (see S1 Text). In CoV-infected cells, vRNA synthesis entails not only genome replication but also the production of a nested set of subgenomic RNAs encoding the structural and so-called "accessory" viral proteins. In terms of RNA copy number, genomic RNA represents only a small fraction of the total vRNA [21,37], but because of its much larger size, the relative amount of label incorporated per genome copy is significantly higher than for subgenomic mRNAs. For example, in MERS-CoV-infected cells, genomic RNA constitutes about 4% of the vRNA molecules [21]) but should incorporate 32% of the [$^3$H]uridine label when compensating for its size and somewhat higher relative uracil content.

A key advantage of our approach over the use of modified precursors (e.g., Br-uridine) is that detection of the label does not rely on immunolabeling. This makes autoradiography

compatible with high-contrast EM sample preparation protocols that provide excellent morphology at the price of epitope integrity. Moreover, as the signal derives from radioactive disintegrations, EM autoradiography is a very sensitive technique, which, in principle, should allow for short labeling pulses, essential to minimize the chance of migration of labeled vRNA products from their site of synthesis (Fig 4). Nevertheless, the pulse should be long enough to enable the internalization of the tritiated uridine, its conversion into $^3$H-UTP in the cell, and its incorporation into vRNA. In order to explore the practical limits of the approach, we tested different labeling pulses in Vero E6 cells infected with SARS-CoV, measuring the amount of radioactive label incorporated into RNA (Fig 4A). Although minimal labeling occurred within the first 10 minutes, a sharp increase in signal was observed between 10 and 20 minutes after label administration. Consequently, only samples labeled for 20 minutes or longer were prepared and processed for EM autoradiography analysis.

Abundant autoradiography signal was detected by EM in SARS-CoV-infected cells pulse-labeled for 20 minutes (Fig 4B). The signal accumulated in the regions that contained virus-induced membrane modifications, which aligned with the idea that these structures are the primary platforms for vRNA synthesis. This largely accepted notion, however, does not formally exclude the possibility that vRNA synthesis could also be associated with other cellular membranes, albeit to a lower extent. Such an association with morphologically intact membranes could be important, for example, in the first stages of infection, when the levels of the viral membrane-remodeling proteins are still low.

Importantly, establishing the association of autoradiography signal with specific subcellular structures requires a detailed quantitative analysis, as the autoradiography signal can spread up to a few hundred nanometers from the original radioactive source (see S1 Text). This type of analysis was used to compare CoV- and mock-infected cells labeled for different periods of time. To this end, we analyzed the autoradiography signal present in hundreds of regions that were randomly picked from large EM mosaic images [38] and calculated labeling densities and relative labeling indexes (RLIs) per compartment [39] (see Materials and methods, S3 Data). The results for SARS-CoV did not show association of vRNA synthesis with any subcellular structure other than ROs (Fig 4C). In infected cells labeled for a short period of time (20 minutes), the RO labeling densities were 1 order of magnitude higher than for any other subcellular structure. Even though the dispersion of signal around the radioactive source would inevitably cause "signal leakage" from these active ROs to neighboring organelles like the ER, none of those alternative locations showed an RLI significantly higher than 1 (RLI $\leq$ 1 indicates unspecific labeling), and their labeling densities were comparable to those in the control mock-infected cells. An increase in labeling densities in some subcellular regions was observed in infected cells when the labeling time was extended to 60 minutes. This could be explained by signal leakage in combination with the migration of vRNA from its site of synthesis, possibly toward the site of virus assembly on membranes of the ER-Golgi intermediate compartment (ERGIC) [40–42]. Indeed, virion-containing regions showed the sharpest increase in labeling density when extending the labeling time. Similar results followed from the analysis of MERS-CoV-infected Huh7 cells (Fig 4D), although no clear signs of RNA migration were observed in this case. Taken together, these results suggest that CoV RNA synthesis is restricted to the RO regions of the infected cell.

## DMVs are the primary site of coronaviral RNA synthesis

Our next goal was to determine which elements of the coronaviral RO (DMVs, CM/zippered ER, and/or DMSs) are directly involved in vRNA synthesis. A first answer to this question became readily apparent as regions containing DMVs, but not either of the other RO structural

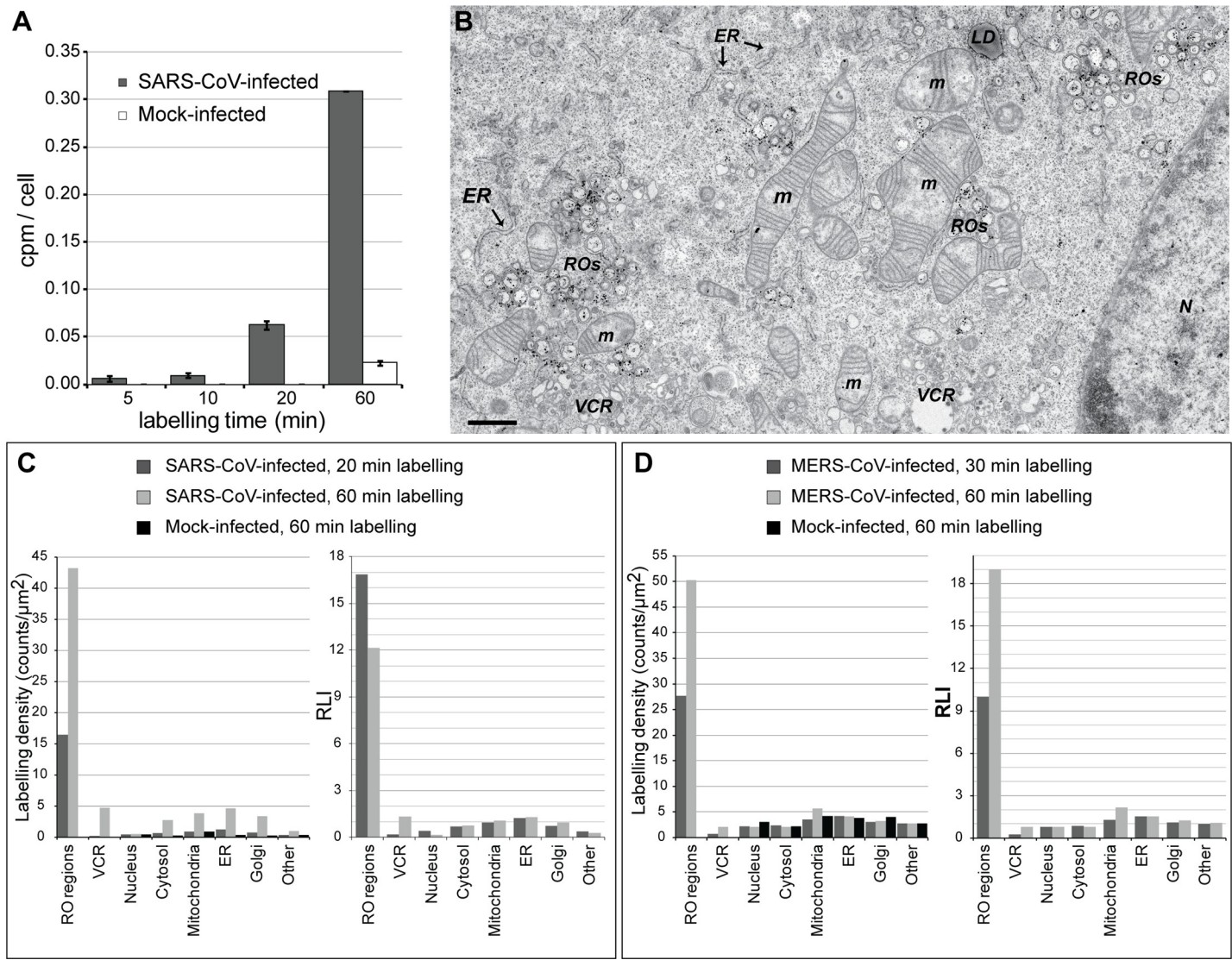

**Fig 4. CoV RNA synthesis is confined to RO regions.** Newly synthesized vRNA was metabolically labeled by providing tritiated uridine to CoV-infected cells pretreated with actinomycin D to limit host transcription. (A) Analysis of the amount of radioactive label incorporated into RNA as a function of the labeling time in SARS-CoV-infected Vero E6 cells (MOI 10), as measured by scintillation counting on the RNA isolated from the cells (underlying numerical data in S2 Data). The label was provided simultaneously to all the samples at 6 hpi. (B-D) EM detection by autoradiography. (B) Overview of a SARS-CoV-infected Vero E6 cell (MOI 10, 7 hpi, labeled for 20 minutes). Autoradiography grains accumulate in the RO regions. Scale bar, 1 μm. (C, D) Quantification of the autoradiography signal per subcellular structure (see also S3 Data). Labeling densities and RLIs in different subcellular regions of (C) Vero E6 cells infected with SARS-CoV (MOI 10) or (D) Huh7 cells infected with MERS-CoV (MOI 5). Radioactively labeled uridine was provided for the indicated periods of time immediately before fixation at 7 hpi and 12 hpi, respectively. These time points represent, respectively, the middle (SARS-CoV) or late (MERS-CoV) exponential phase of viral replication [21,34]. Control mock-infected cells are excluded from the RLI plots, as RLI comparisons between conditions require the same number of classes (subcellular regions) and these cells lack ROs and virions. CM, convoluted membranes; CoV, coronavirus; cpm, counts per minute; DMS, double-membrane spherule; DMV, double-membrane vesicle; EM, electron microscopy; ER, endoplasmic reticulum; HCoV-229E, human coronavirus 229E; hpi, hours postinfection; IBV, infectious bronchitis virus; LD, lipid droplet; m, mitochondrion; MERS-CoV, Middle East respiratory syndrome-CoV; MHV, murine hepatitis virus; MOI, multiplicity of infection; N, nucleus; RLI, relative labeling index;; RO, replication organelle; SARS-CoV, severe acute respiratory syndrome-CoV; VCR, virion-containing region; vRNA, viral RNA.

elements, were densely labeled in cells infected with SARS-CoV (Fig 4B) or MERS-CoV (Fig 5). Interestingly, not all the DMV clusters in a sample, and sometimes even within a cell, appeared equally densely labeled, which suggests that the levels of vRNA synthesis in DMVs are variable and can change in time. The active role of DMVs in vRNA synthesis in MERS-CoV-infected cells was further corroborated by a detailed analysis of the distribution of

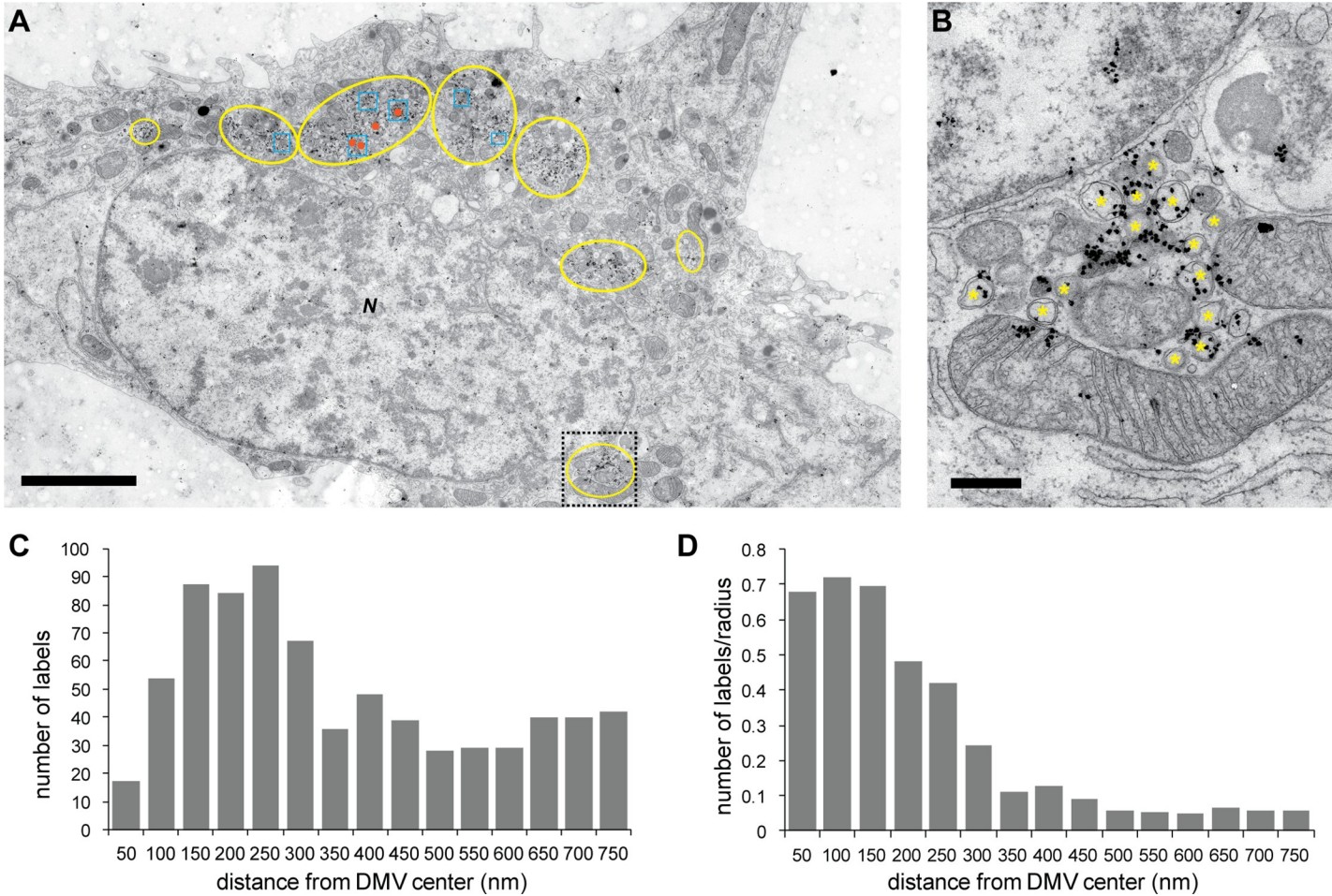

**Fig 5. DMVs are sites of vRNA synthesis.** Analysis of the association of autoradiography signal with DMVs in MERS-CoV-infected Huh7 cells (MOI 5). The cells were pretreated with actinomycin D at 10 hpi and labeled with tritiated uridine for 30 minutes immediately before fixation (12 hpi). (A) Overview of an infected cell in which regions with different virus-induced modifications are annotated in yellow (DMVs), blue (CM), and orange (DMSs). Several densely labeled regions containing DMVs (but not the other virus-induced structures) are apparent. A close-up of one of these regions (boxed area) is shown in (B), with DMVs highlighted by yellow asterisks. (C, D) Distribution of the autoradiography signal around DMVs ($n_{DMVs}$ = 36, see Materials and methods for selection criteria and details, and S4 Data for the underlying numerical data). The data are plotted (C) as a histogram or (D) normalized by the radius to the DMV center to account for the increase in the perimeter of the screened area with the distance. Scale bars, (A) 5 μm, (B) 500 nm. CM, convoluted membranes; DMS, double-membrane spherule; DMV, double-membrane vesicle; hpi, hours postinfection; MERS-CoV, Middle East respiratory syndrome-coronavirus; MOI, multiplicity of infection; N, nucleus; vRNA, viral RNA.

autoradiography signal around isolated DMVs (Fig 5C and 5D). In case of a random distribution (i.e., if the DMVs were not a true radioactive source), the number of autoradiography grains around the DMVs would simply increase with the distance, as the perimeter of the screened area also increases and with it, the chances of detecting background signal. The signal around DMVs was clearly not random, showing maximum levels in the proximity of these structures (Fig 5C). Moreover, the distribution normalized by the distance made apparent a maximum around the average radius of the DMVs analyzed (133 ± 28 nm, $n$ = 36) (Fig 5D), aligning with the idea that vRNA synthesis takes place in membrane-bound enzymatic complexes.

Next, we specifically investigated the possible involvement in vRNA synthesis of CM and/ or DMSs (Fig 6), which were always present in membrane-modification clusters that also contained DMVs. A close inspection of multiple CM showed that these structures were mainly devoid of signal and that the occasional silver grains present primarily appeared in the

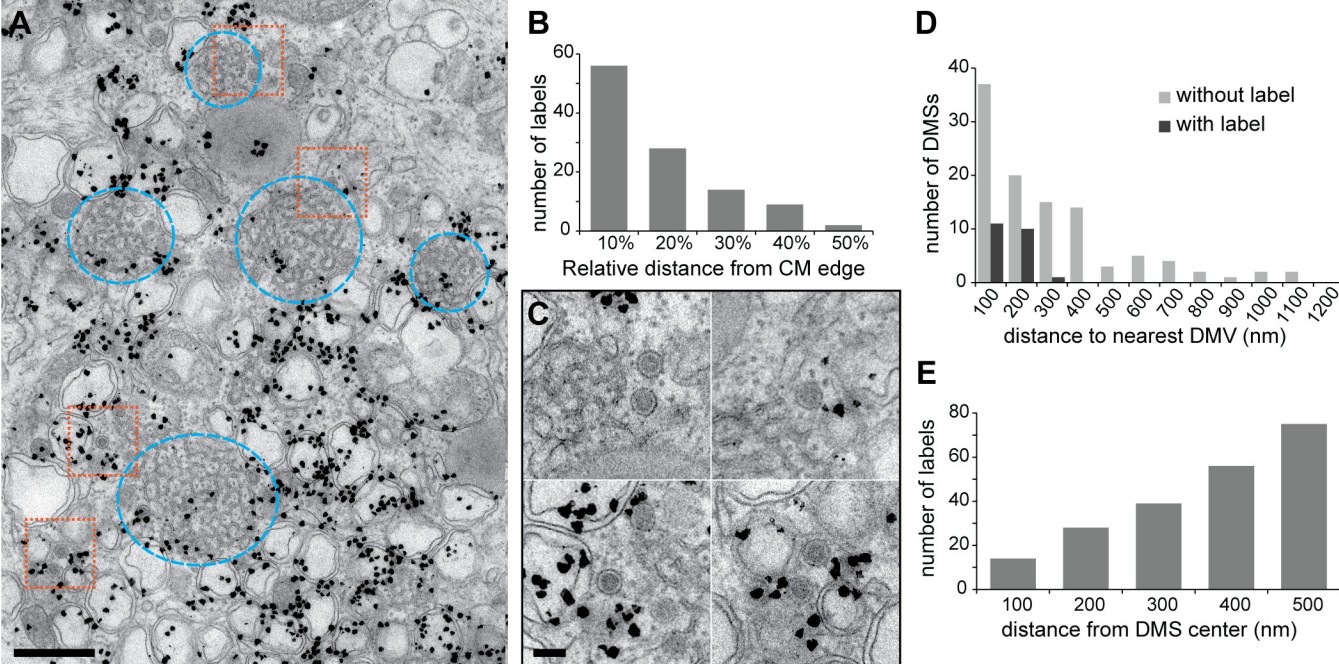

**Fig 6. Newly synthesized vRNA signal does not clearly associate with CM or DMSs.** (A) Overview of a cluster of MERS-CoV-induced membrane modifications in Huh7 cells prepared as described in Fig 5. Some DMSs are boxed in orange, and regions with CM are encircled in blue. In comparison with the densely labeled surrounding DMVs, these regions are relatively devoid of autoradiography signal. (B) The distribution of autoradiography grains on CM was not homogeneous ($n_{CM} = 9$), and label was predominantly found close to the boundaries of the CM, as expected if the signal arises from the surrounding DMVs. (C-E) Analysis of the label around/on the DMSs (see Materials and methods for selection criteria and details). (C) Enlargements of the DMS areas boxed in (A). Most DMSs were devoid of signal, and those who contained label were close to labeled DMVs (D) ($n_{DMS} = 127$). (E) The distribution of signal around DMSs shows an increase in the amount of autoradiography grains with the distance from the DMS center, as expected from a random distribution ($n_{DMSs} = 58$). The underlying numerical data for the plots are in S5 Data. Scale bars, (A) 500 nm, (C) 100 nm. CM, convoluted membranes; DMS, double-membrane spherule; DMV, double-membrane vesicle; vRNA, viral RNA.

periphery of the CM, therefore likely stemming from surrounding labeled DMVs (Fig 6A and 6B). Similar observations were made for DMSs (Fig 6C–6E): most of them (83%) lacked signal, and the rest was close to abundantly labeled DMVs. Furthermore, no signs of DMSs acting as a signal source were apparent in the distribution of signal around them, which resembled that of a random pattern (Fig 6E, compare with Fig 5C).

To explore whether these observations could be extended to distantly related CoVs, we expanded this type of analysis to the gamma-CoV IBV. IBV-induced DMSs are particularly abundant, and a large proportion of them have an open configuration, which contributed to the hypothesis that these open DMSs could be engaged in vRNA synthesis [26]. However, no evidence supporting this hypothesis could be derived from our detailed analysis of the autoradiography signal in IBV-infected cells, which essentially produced the same results as for MERS-CoV (Fig 7). Taken together, our observations clearly point at coronaviral DMVs as the active site of vRNA synthesis and seem to indicate that neither CM/zippered ER nor DMSs are effectively involved in this process, at least not to a significant level above the detection limits of our method.

## Viral markers in the MERS-CoV RO

If CM and DMSs are not involved in vRNA synthesis, what is the role (if any) of these RO structural elements in CoV replication? To investigate this, we analyzed the subcellular

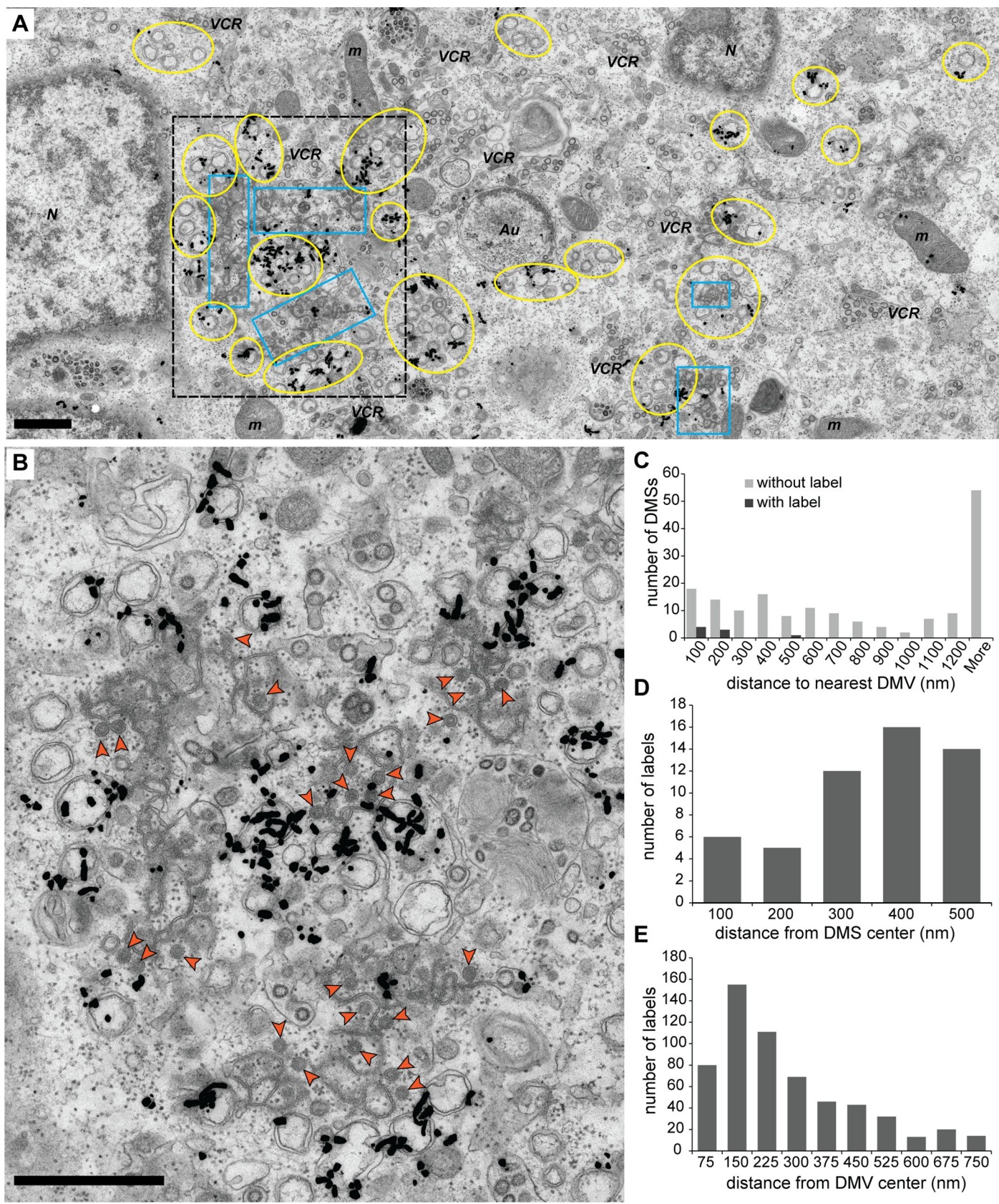

**Fig 7. Metabolic labeling of newly synthesized vRNA in IBV-infected cells and analysis of the autoradiography signal.** Vero cells infected with IBV were pretreated with actinomycin D for 1 hour, then labeled for 30 or 60 minutes with tritiated uridine, immediately fixed at 16 and 17 hpi, respectively, and processed for autoradiography EM. These time points allow for a second cycle of infection and were chosen to increase the number of infected cells (see Materials and methods). (A) Overview of an IBV-infected Vero cell labeled for 60 minutes. The areas containing DMVs and zippered ER are outlined in yellow and blue, respectively, and other subcellular structures are also annotated. The autoradiography signal accumulates in areas of virus-induced membrane modifications that often only contain DMVs, in alignment with DMVs having an active role in vRNA synthesis. (B) Close-up of the area boxed in black in (A), which contains DMVs, zippered ER and DMSs (orange arrowheads). The contrast between the densely labeled DMVs and the zippered ER and DMSs largely lacking signal is apparent and suggests that the autoradiography grains sometimes present on the latter structures arose from radioactive disintegrations in the surrounding active DMVs. (C) In agreement with this possibility, most of the DMSs (96%) were devoid of signal, and most of those that contained label where close to an active DMV ($n_{DMS}$ = 178). (D) Furthermore, the distribution of autoradiography grains around DMSs resembled that of a random distribution, in which the number of grains increase with the distance ($n_{DMS}$ = 106). (E) In contrast, a similar analysis of the signal around the DMVs proved that these structures are associated with vRNA synthesis, as the signal reaches maximum values in the proximity of the DMVs ($n_{DMVs}$ = 106). (C, D) See Materials and methods for the selection criteria and details and S6 Data for the underlying numerical data. Scale bars, 1 μm. Au, autophagosome; DMS, double-membrane spherule; DMV, double-membrane vesicle; EM, electron microscopy; ER, endoplasmic reticulum; hpi, hours postinfection; IBV, infectious bronchitis virus; m, mitochondrion; N, nucleus; VCR, virion-containing region; vRNA, viral RNA.

location of different viral markers in MERS-CoV-infected cells by immunoelectron microscopy (IEM) (Fig 8).

CoV-induced DMSs, with their electron-dense content and their remarkably regular size, are particularly intriguing structures. Revealing DMSs in IEM samples, however, was challenging and, in our hands, required a modified protocol for the preparation of thawed cryo-sections [43] that, unfortunately, failed to make DMVs apparent (see Materials and methods). Given the similar size of DMSs and virus particles, we first considered the possibility that the DMSs would represent some kind of nonproductive virus assembly event on the CM. Although new CoV particles typically assemble in the ERGIC [40–42,44–45], virus budding from ER membranes, from which CM originate, can also occur [40,46–47], and we regularly observed it in MERS-CoV- and IBV-infected cells (e.g., Fig 2B). To investigate this possibility, we used antibodies against several of the structural proteins, namely, the nucleocapsid protein N, the envelope membrane protein M, and the spike protein S. As expected, all of them were detected in newly formed MERS-CoV particles present in budding vesicles (Fig 8A–8C). The M and S proteins also localized to the Golgi complex, aligning with previous observations for other CoVs [20,44,48–49]. The MERS-CoV N protein was found in regions with CM and DMSs, though the distribution of signal was homogenous and DMSs were not particularly densely labeled (Fig 8D). The presence of the N protein in the viral RO has also been shown for MHV [20] and suggested by a number of colocalization studies [42,50–53] and may be related to a possible role in vRNA synthesis of this multifunctional protein [54]. Importantly, neither DMSs nor CM labeled for the M protein (the most abundant viral envelope protein and the presumed orchestrator of virion assembly) or the S protein (Fig 8E and 8F).

Previously, the CM induced by SARS-CoV and MHV were shown by IEM to accumulate viral nsps, whereas dsRNA signal was primarily found inside the DMVs [15,20]. Similarly, nsp3 mapped to the CM induced in MERS-CoV infection but also to the DMSs to a comparable extent (Fig 8G). Our attempts to combine dsRNA antibody labeling with thawed cryo-sections were unsuccessful, which made us resort to HPF-FS samples. In these, however, while DMVs were easily detected, the morphology of CM and DMSs was less clearly defined. Nevertheless, dsRNA signal was clearly associated with DMVs, whereas the dark membranous regions between DMVs that we interpreted as CM and DMSs clusters appeared devoid of signal (Fig 8H and 8I).

In summary, for the antibodies tested (recognizing N, M, S, nsp3, and dsRNA), the labeling pattern in MERS-CoV-induced DMSs closely resembled that of the CM, from which they seem to derive. The absence of labeling for key proteins in virus assembly, like the M and S proteins, strongly suggest that DMSs do not represent (spurious) virus assembly events.

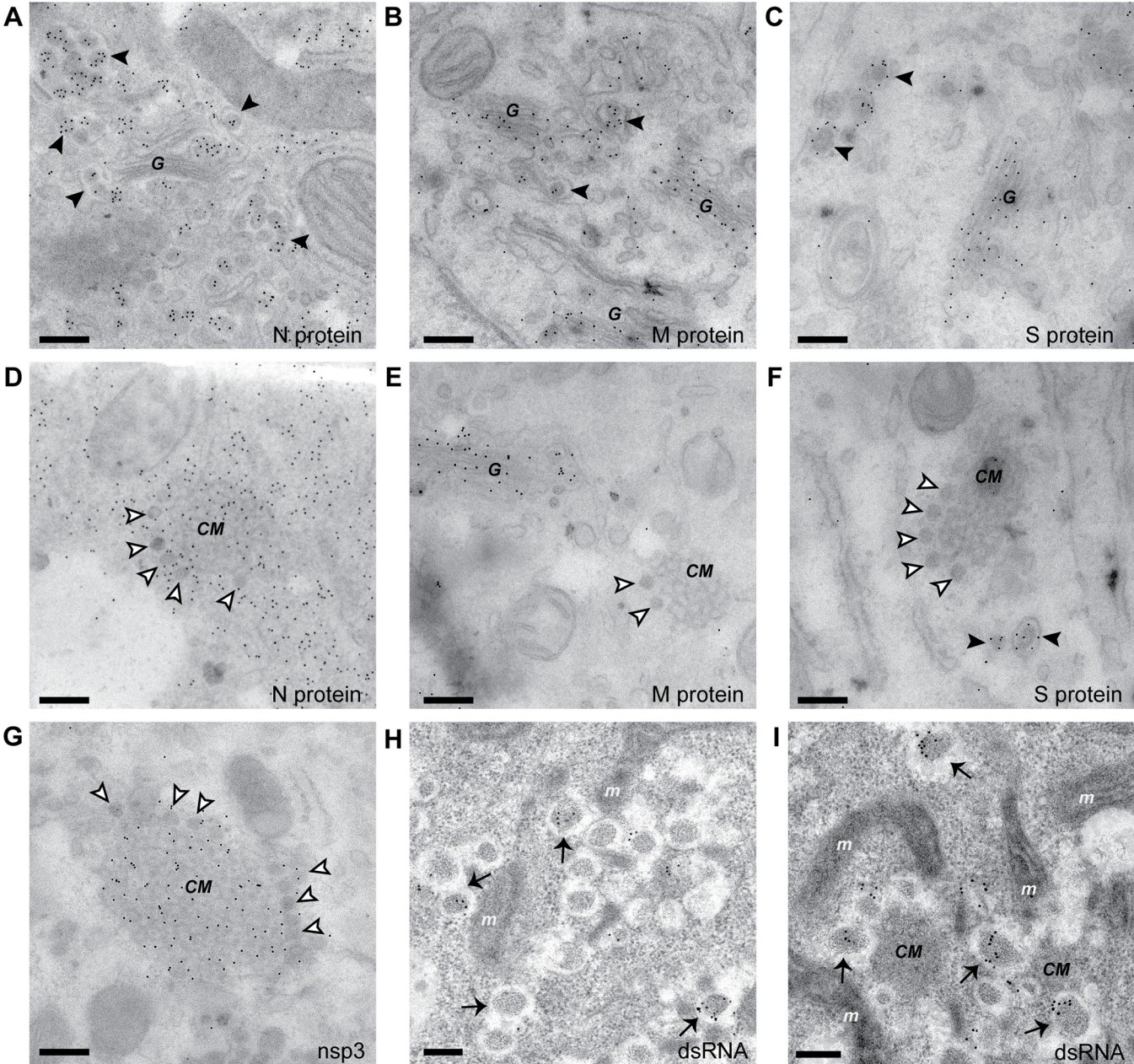

**Fig 8. IEM detection of viral markers in MERS-CoV-infected cells.** (A-G) Immunogold labeling of thawed cryo-sections of MERS-CoV-infected Huh7 cells (12 hpi) for the detection of the indicated viral proteins. (A-C) Structural proteins were detected on virions (black arrowheads) and, for the M and S proteins, also on Golgi cisterna. While regions containing DMS (white arrowheads) and CM labeled for the N protein (D) and nsp3 (G), the M and S protein were not detected in these areas. (H-I) Immunogold labeling of dsRNA in HPF-FS samples of MERS-CoV-infected Huh7 cells (13 hpi). The label accumulated on DMVs, which could be easily detected in this type of samples (black arrows), whereas the regions with CM and DMSs, which appeared as dark areas among the DMV clusters, were devoid of dsRNA signal. Scale bars, 250 nm. CM, convoluted membranes; DMS, double-membrane spherule; dsRNA, double-stranded RNA; G, Golgi complex; HPF-FS, high-pressure freezing, freeze-substitution; hpi, hours postinfection; IEM, immunoelectron microscopy; m, mitochondrion; MERS-CoV, Middle East respiratory syndrome-CoV; nsp3, nonstructural protein 3.

## Discussion

The comprehensive analysis presented here demonstrates that viruses across different CoV genera induce essentially the same type of membrane structures. After somewhat disparate

observations [15,20–21,26–27,47], the unifying model that emerges from our study is that of a CoV RO comprising 3 basic types of double-membrane structural elements: (1) CM or zippered ER, which would represent branched or unbranched configurations of paired-ER membranes; (2) small DMSs that appear to arise from CM or zippered ER; and (3) DMVs. These structural elements are largely interconnected and connected to the ER, together forming the reticulovesicular network that is typical of the coronaviral RO, including, in all likelihood, the RO of SARS-CoV-2, a close relative of SARS-CoV.

Our results clearly confirm and extend DMVs as the primary—if not only—site of vRNA synthesis within the coronaviral RO. This point has been subject to quite some speculation, due in part to the limited experimental data directly addressing this question. An early study using Br-UTP and IEM to detect newly synthesized vRNA mapped signal in DMV regions of MHV-infected cells [14], but the poor preservation of IEM samples did not allow the recognition of CM and DMSs, typically present in these regions. Later, a light microscopy study using 5-ethynil uridine labeling and click chemistry suggested that vRNA synthesis could take place, at least partially, in a different location than the DMVs [55]. Alternative interpretations of those results, like migration of vRNA from the DMVs during the relatively long labeling times used (60 minutes) appear now more plausible. Several other studies further contributed to question the idea of DMVs being the sites of vRNA synthesis by showing that higher numbers of DMVs did not necessarily translate into larger amounts of vRNA or provide a competitive advantage [24–25,56]. It should be noted, however, that the number of DMVs may not necessarily correlate with the number of active replication complexes that they contain at a given time, something that is also suggested by our observed variations in the level of autoradiography signal among DMVs in the same sample.

While the finding of open DMSs, similar to the invaginations that many +RNA viruses use as replication sites, made them attractive candidate sites for vRNA synthesis [26,57], we could not detect vRNA synthesis associated with them, nor with CM or any other subcellular structure using a highly sensitive technique like autoradiography. Although some level of vRNA synthesis in any of these structures cannot be completely discarded, our results suggest that, if present, this would only be marginal compared with the abundant DMV-associated activity.

Given the prevalence of CM and DMSs across different CoV genera, it is tempting to speculate that these structures must play a role in virus replication. In particular, for the highly regular DMSs, it is hard to imagine that they would lack a specific function; however, their role remains elusive. A suggestive possibility was that DMSs represent nonproductive virus assembly events, but the lack of DMS labeling for key structural proteins seems to rule out this option. In fact, for the viral markers tested, no differences were apparent in the labeling patterns of DMSs and the CM from which they seem to derive. Although their distinct morphology implies that DMSs likely contain specific host or viral proteins, these factors—which may give important clues about DMS function—remain to be identified. Another possibility is that CM and DMSs represent DMV precursors. In this scenario, ER membrane pairing would first give rise to CM, which would then produce DMSs that could eventually expand into DMVs. Pairing of ER membranes, which appears to be driven by 2 key transmembrane viral proteins (nsp3 and nsp4), indeed, is the likely first step in CoV DMV biogenesis, as suggested by studies in cells ectopically expressing these proteins [22–23,27,58]. CM, however, seem to appear later in infection than DMVs, as documented for SARS-CoV, MHV, and MERS-CoV [15,20–21]. While this may argue against CM and DMSs being DMV precursors, these virus-induced structures could still represent basic membrane-remodeling stages in DMV formation that, later in infection, do not progress adequately. It has been proposed, for example, that CM could be a form of cubic membranes [23,59], which are membrane aggregates resulting from ER protein overexpression [60]. CM, which abundantly label for viral nsps [15,20] and

proliferate late in infection [15,20–21], could thus be a by-product of viral protein overexpression.

Our results add to studies that, in the last years and after much speculation, have started to provide experimental evidence that the DMVs induced by +RNA viruses are active sites of vRNA synthesis [11,61–63]. However, it is not clear that DMVs always play the primary role in virus replication that we demonstrate here for CoV. For picornaviruses, for example, virus-induced single-membrane structures, which are DMV precursors and also active sites of vRNA synthesis, could well be more relevant as they predominate at the peak of vRNA replication [10–11,63]. By clearly pointing to DMVs as the key sites for CoV replication, our results also bring back to center stage some of the challenges that CoV-induced DMVs pose, which extend to the distantly related arterivirus family and probably also to other members of the order *Nidovirales*. In contrast with the DMVs induced by other +RNA viruses [10–11,13,63], the DMVs in nidovirus-infected cells appear to lack membrane openings that would connect their inner compartment with the cytosol to allow import of precursors and export of genomic and subgenomic viral mRNAs [15–16,26]. This topological conundrum, however, starts with the assumption that vRNA synthesis takes place inside the DMVs, yet the evidence so far is insufficient to ascertain this point. Shielding of vRNA inside DMVs arguably provides the most straightforward explanation for the observation that intact membranes protected vRNA from nuclease treatment [64]. However, protection of vRNA could also be achieved in the outer DMV membrane through protein complexes that would rely on membranes for assembly/stability.

Although a definitive answer to this issue is still missing, our results allow narrowing down the possible scenarios. If the DMVs are closed structures lacking an import/export mechanism, vRNA synthesis on the outer DMV membrane appears as a necessity to provide vRNA for translation and encapsidation. Then, vRNA synthesis inside the DMVs would also be required to explain the observed accumulation of (presumably viral) dsRNA. Although this is not at all an attractive possibility, as vRNA synthesis inside the DMVs and even DMV formation would appear spurious, it cannot be completely discarded at this stage. The most appealing scenario is that in which vRNA synthesis only takes place inside the DMVs. This would provide the compartmentalization of vRNA synthesis that may be most beneficial for viral replication, although it would require the existence of a yet unidentified import/export mechanism. Notice that a transport mechanism would also be needed in the third possible scenario, i.e., if vRNA synthesis occurs only on the outer DMV membrane, to account for the accumulation of dsRNA inside the DMVs that could then perhaps serve to hide excess vRNA from detection by innate immune sensors [65].

Despite the lack of clear openings in the membranes of CoV-induced DMVs, a mechanism allowing exchange of material with the cytosol is conceivable. The openings of DMV may be extremely short-lived, and therefore, they may have eluded detection by EM. This may become apparent in the future, if mutant CoV-inducing DMVs with slower dynamics are found. An appealing alternative is the existence of molecular pores that may well be undetectable in conventional EM samples. Precedents of molecular complexes bridging double membranes include the large nuclear pore complex but also small transporters through the mitochondrial or chloroplast membranes. Visualizing such a putative small molecular pore on DMVs is a formidable challenge that would likely require the use of cryo-EM to preserve macromolecular components. Emerging techniques like in situ cryotomography, which allows the visualization of structures in their cellular context at macromolecular resolution, may be key to realize this next step and to understand how CoVs exploit the complex architecture of DMVs.

## Materials and methods

### Cells, viruses, and infections

Huh7 cells (kindly provided by Ralf Bartenschlager, Heidelberg University) were grown in Dulbecco's modified Eagle's medium (DMEM; Lonza) supplemented with 8% (vol/vol) fetal calf serum (FCS; Bodinco), 2 mM L-glutamine (PAA Laboratories), and nonessential amino acids (PAA Laboratories). Vero cells (ECACC 84113001) were cultured in Eagle's minimal essential medium (EMEM; Lonza) with 8% FCS and 2 mM L-glutamine. Vero E6 cells (ATCC CRL-1586) were maintained in DMEM supplemented with 8% FCS. Mouse 17 clone 1 (17Cl1) cells (gift from Stuart Siddell, University of Bristol) were grown in DMEM supplemented with 8% FCS and 8% (vol/vol) tryptose phosphate broth (Life Technologies). Penicillin and streptomycin (90 IU/ml, PAA Laboratories) were added to all media.

The CoVs used in this study include MERS-CoV (strain EMC/2012, [28–29], SARS-CoV (strain Frankfurt-1, [66]), MHV (ATCC VR-764), HCoV-229E ([67]), and IBV (strain Beau-R, [68]). All the infection experiments were carried out at 37˚C, except for HCoV-229E infections, which were performed at 33˚C. Cells were infected at high MOI (5–10), with the exception of IBV (MOI < 1). Times postinfection in the middle or late exponential phase of viral replication [21,34–36] were selected for analysis to favor both a good amount of label incorporation and a variety and abundant presence of virus-induced membrane structures. In the case of IBV, however, because of the low titer and to increase the number of infected cells, the times postinfection (16–17 hpi) extended beyond the first cycle of viral replication infection. For MHV infections, 1 μM HR2 peptide was added to the cell medium to prevent syncytia formation [69]. Control mock-infected cells were included in all the experiments. Infections were routinely assessed by immunofluorescence assays on parallel samples, essentially processed as previously described [51]. All work with live SARS-CoV and MERS-CoV was performed inside biosafety cabinets in the biosafety level 3 facility at Leiden University Medical Center.

### Antibodies

For IEM of MERS-CoV-infected cells, the antibodies used included a previously described rabbit antiserum that recognizes SARS-CoV nsp3 protein and cross-reacts with MERS-CoV nsp3 [21,70], a polyclonal rabbit antibody generated against full-length MERS-CoV N protein (Sino Biological), and a mouse monoclonal antibody (J2) specific for dsRNA [71], purchased from Scicons. A MERS-CoV M-specific rabbit antiserum was ordered from Genscript and produced using as antigen a synthetic peptide representing the C-terminal 24 residues of the protein (CRYKAGNYRSPPITADIELALLRA). The specificity of the antiserum was verified by western blot analysis and immunofluorescence microscopy using samples from MERS-CoV-infected Vero or Huh7 cells, as described previously [21], while using preimmune serum and mock-infected cell lysates as negative controls. The human monoclonal antibody used against MERS-CoV spike protein was kindly provided by Dr. Berend Jan Bosch (Utrecht University) and has been described elsewhere as 1.6f9 [72].

### Metabolic labeling and label incorporation measurements

To label newly synthesized RNA, CoV-infected cells and control mock-infected cells were incubated for different periods of time with tritiated uridine ([5-$^3$H]uridine, 1 mCi/ml, Perkin Elmer), which was mixed in a 1:1 ratio with double-concentrated medium. Cellular transcription was blocked by providing the cells with 10 μg/ml actinomycin D both during labeling and in a preincubation step of 1 to 2 hours, depending on the specific set of samples. Directly after labeling, the cells were extensively washed with phosphate-buffered saline (PBS) and

immediately fixed for EM autoradiography. A parallel set of samples was included in every experiment for label incorporation measurements. These cells were lysed using TriPure Isolation Reagent (Roche), and the RNA was isolated following the manufacturer's instructions. The incorporation of radioactive label into RNA was then measured using scintillation counting.

## Sample preparation for electron microscopy

For ultrastructural analysis, autoradiography, and tomography, EM samples of CoV-infected cells were prepared by chemical fixation. Chemical fixation was chosen over the alternative of HPF-FS to increase the yield of cells per EM grid and thus facilitate the autoradiography quantitative analysis, as it was observed that infected cells were easily washed away and lost during FS. At the desired times postinfection, the cells were fixed for 30 minutes with 1.5% (vol/vol) glutaraldehyde (GA) in 0.1 M cacodylate buffer (pH 7.4). Cells infected with SARS- or MERS--CoV were further maintained in the fixative overnight. After fixation, the samples were washed with 0.1 M cacodylate buffer either once or, in samples destined to autoradiography, 3 times (5 minutes each) to favor the elimination of unincorporated [5-$^3$H]uridine. Next, the samples were treated with 1% (wt/vol) $OsO_4$ at 4˚C for 1 hour, washed with 0.1 M cacodylate buffer and Milli-Q water, and stained at room temperature for 1 hour with 1% (wt/vol) low-molecular weight tannic acid (Electron Microscopy Science) in 0.1 M cacodylate buffer (SARS-CoV autoradiography experiments), or with 1% (wt/vol) uranyl acetate in Milli-Q (rest of the samples). Following a new washing step with Milli-Q water, the samples were dehydrated in increasing concentrations of ethanol (70%, 80%, 90%, 100%), embedded in epoxy resin (LX-112, Ladd Research), and polymerized at 60˚C. Sections were collected on mesh-100 copper EM grids covered with a carbon-coated Pioloform layer, and poststained with 7% (wt/vol) uranyl acetate and Reynold's lead citrate.

**EM autoradiography samples.** A step-by-step protocol for the preparation of autoradiography samples can be found in http://dx.doi.org/10.17504/protocols.io.bfrtjm6n. In short, EM grids with ultrathin cell sections (50-nm thick) were first attached to glass slides, including in each glass slide a grid for every condition tested within a given experiment, to be later developed simultaneously. In a dark room, a thin layer of nuclear emulsion ILFORD L4 was placed on top of the grids with the help of a wire loop [32]. Samples were maintained in the dark in a cold room for several weeks until development, which was performed as described in [73]. The progress in the exposure of the nuclear emulsion to radioactive disintegrations was evaluated regularly by EM until the number of autoradiography grains was sufficient for analysis (approximately 6 weeks for SARS-CoV-infected cells, 21 weeks for MERS-CoV-infected cells, and 12 to 13 weeks for IBV-infected cells).

**IEM.** Several types of MERS-CoV-infected cell samples were prepared for IEM. The labeling of viral proteins was performed on thawed cryo-sections, which are optimal for epitope preservation. To this end, at 12 hpi, infected and mock-infected cells were first chemically fixed for 1 hour at room temperature with 3% (wt/vol) paraformaldehyde (PFA) and 0.25% (vol/vol) GA in 0.1 M PHEM buffer (60 mM PIPES, 25 mM HEPES, 10 mM EGTA, 2 mM $MgCl_2$ [pH 6.9]), briefly washed with 0.1 M PHEM buffer, and stored at 4˚C in 3% (wt/vol) PFA in 0.1 M PHEM buffer until transfer from the biosafety level 3 facility for further processing. To prepare EM samples, the cells were first pelleted and embedded in 12% (wt/vol) gelatin. Cubes of approximately 1 mm$^3$ in size were cut from these pellets, infiltrated with 2.3 M sucrose for cryo-protection, snap-frozen in liquid nitrogen, and sectioned by cryo-ultramicrotomy. Thawed cryo-sections (70-nm thick) deposited on EM grids were incubated first with the corresponding primary antibody and then with protein A coupled to colloidal 10-nm gold

particles. Virus-induced membrane modifications, however, were not discernible in these samples, and clear signs of membrane extraction were present. To tackle this issue, we used a previously described modification of the protocol that includes sequential poststaining steps with 1% (wt/vol) OsO4, 2% (wt/vol) uranyl acetate, and Reynold's lead citrate after immuno-gold labeling and prior to the final embedding in a thin layer of 1.8% methyl cellulose [43]. Both CM and DMSs were clearly recognizable in these samples, whereas DMVs were still not apparent and may have been extracted, as empty areas were often found in the vicinity of CM regions.

The detection of dsRNA required the preparation of HPF-FS samples. For biosafety considerations, Huh7 cells grown on sapphire disks and infected with MERS-CoV were first fixed overnight at 13 hpi with 3% (wt/vol) PFA and 0.25% (vol/vol) GA in 0.1 M PHEM buffer. Then, the samples were frozen with a Leica EM PACT2, after which they were freeze-substituted in a Leica AFS2 system with 0.1% (wt/vol) uranyl acetate as previously described [74], with the only modification that acetone was replaced by ethanol from the last washing step before Lowycril infiltration onwards. Cell sections (75-nm thick) were incubated with the primary mouse antibody, then with a bridging rabbit anti-mouse-IgG antibody (Dako Cytomation), and finally with protein A coupled to 15-nm gold particles. After immunolabeling, samples were additionally stained with 7% (wt/vol) uranyl acetate and Reynold's lead citrate.

**Electron microscopy imaging.**   Individual 2D-EM images were acquired in a Tecnai12 BioTwin or a Twin electron microscope, equipped with an Eagle 4k slow-scan change-couple device (CCD) camera (Thermo Fisher Scientific [formerly FEI]) or a OneView 4k high-frame rate CMOS camera (Gatan), respectively. Mosaic EM images of large grid areas were generated for the quantitative analysis of autoradiography samples, using overlapping automatically collected images (pixel size 2 nm, Tecnai 12 BioTwin) that were subsequently combined in a composite image as described in [38].

**Electron tomography.**   Prior to the poststaining step, semithin sections (150-nm thick) of CoV-infected cells were incubated with protein A coupled to 10-nm colloidal gold particles that served later as fiducial markers for alignment. Dual-axis tilt series were collected in a Tecnai12 BioTwin microscope using Xplore 3D acquisition software (Thermo Fisher Scientific), covering each 120˚–130˚ around the specimen with an angular sampling of 1˚ and a pixel size of 1.2 nm. The alignment of the tilt series and tomogram reconstruction by weighted back-projection was carried out in IMOD [75]. The diameters of DMVs, DMSs, and virions were measured at their equator. DMV profiles, often only roughly circular, were measured over their longest and shortest axes, and the diameter was estimated as the geometric mean of the 2 values. For visualization purposes, the tomograms were first mildly denoised and then processed in Amira 6.0.1 (Thermo Fisher Scientific) using a semiautomatic segmentation protocol as previously described [10].

**Quantitative analysis of autoradiography samples.**   Large mosaic EM maps containing dozens of cell profiles were used for the quantitative analysis of the newly synthesized RNA autoradiography signal (see S3 Data). For each CoV, different conditions (infected and mock-infected cells, plus different labeling times) were compared using only samples developed after the same period of time. The analysis of the signal in different subcellular regions was carried out using home-built software. Areas of 4 $\mu m^2$ were randomly selected from the mosaic EM maps, and the autoradiography grains present in those areas were manually assigned to the underlying cellular structures. The abundance of the different types of subcellular structures was estimated through virtual points in a 5×5 lattice superimposed to each selected area, which were also assigned to the different subcellular classes. Regularly along the process, the annotated data per condition were split into 2 random groups, and the Kendall and Spearman coefficients, which measure the concordance between 2 data sets [76], were calculated. New

random regions were added until the average Kendall and Spearman coefficients resulting from 10 random splits were higher than 0.8 and 0.9, respectively (maximum value, 1). Labeling densities and RLIs were then calculated from the annotated points [39].

For the analysis of the association of vRNA synthesis with each of the different ROs motifs, the specific DMVs, DMSs, and CM included in the analysis were carefully selected. Only individual DMVs that were at least 1 μm away from any other virus-induced membrane modification were selected. For every grain present in an area of 750-nm radius around each DMV, the distance to the DMV center was measured. In the case of DMSs, which were always part of clusters of virus-induced membrane structures, only DMSs in the periphery of these clusters were selected. The quantified signal was limited to subareas devoid of other RO motifs, which were defined by circular arcs (typically 30° to 100°, radius 500 nm) opposite to the RO clusters. CM are irregular structures that appear partially or totally surrounded by DMVs. Only large CM (>0.6 μm across) were selected in order to make more apparent (if present) any decay of the autoradiography signal as the distance to the surrounding DMVs increased. For each autoradiography grain, both the distance to the closest CM boundary ($d_1$) and the distance to the opposite CM edge ($d_2$) were measured. The relative distance to the CM edge was then calculated as $d_1/(d_1+d_2)$ and expressed in percentages. All the measurements in different DMVs, DMSs, and CM were made using Aperio Imagescope software (Leica) and pooled together into 3 single data sets.

## Supporting information

**S1 Video. Electron tomography of the membrane structures induced in MERS-CoV infection.** Animation illustrating the tomography reconstruction and model presented in Fig 1B. The video first shows the tomographic slices (1.2-nm thick) through the reconstructed volume and then surface-rendered models of the different structures segmented from the tomogram: DMSs (orange), CM (blue), and DMVs (yellow and lilac, outer and inner membranes, respectively), ER (green), and a vesicle (silver) containing virions (pink). The movie highlights the DMS association with CM, which, in turn, connect to ER membranes, and these to DMVs. CM, convoluted membranes; DMS, double-membrane spherule; DMV, double-membrane vesicle; ER, endoplasmic reticulum; MERS-CoV, Middle East respiratory syndrome-coronavirus.
(MP4)

**S1 Fig. Detection of DMSs in cryo-fixed and FS samples of CoV-infected cells.** Analysis of previously described samples of CoV-infected cells, prepared for EM either by HPF (A) or cryo-plunging (B). A targeted search revealed the presence of DMSs (white arrowheads) in close association with CM. In comparison with the chemically fixed samples used in this study, the superior ultrastructural preservation of cryo-fixation results in less distorted membranes, but also in a denser cytoplasm and darker CM that makes DMS less apparent. (A) Example from a MERS-CoV-infected Vero cell (16 hpi) in a sample used in [21]. (B) Region in a SARS-CoV-infected Vero E6 cell (8 hpi), adapted from [15]. Scale bars, 250 nm. CM, convoluted membranes; CoV, coronavirus; DMS, double-membrane spherule; EM, electron microscopy; FS, freeze-substituted; HPF, high-pressure freezing; hpi, hours postinfection; MERS-CoV, Middle East respiratory syndrome-coronavirus; SARS-CoV, severe acute respiratory syndrome-CoV.
(TIF)

**S1 Text. Autoradiography in electron microscopy.**
(DOCX)

**S1 Data. Relative sizes of DMVs, DMSs, and virions.** DMS, double-membrane spherule; DMV, double-membrane vesicle.
(XLSX)

**S2 Data. Incorporation of radioactive label into RNA with different labeling times in SARS-CoV-infected cells.** SARS-CoV, severe acute respiratory syndrome-coronavirus.
(XLSX)

**S3 Data. Quantitative analysis of the autoradiography signal per subcellular structure in SARS-CoV and MERS-CoV-infected cells.** MERS-CoV, Middle East respiratory syndrome-coronavirus; SARS-CoV, severe acute respiratory syndrome-coronavirus.
(XLSX)

**S4 Data. Autoradiography signal distribution around DMVs in MERS-CoV-infected cells.** DMV, double-membrane vesicle; MERS-CoV, Middle East respiratory syndrome-coronavirus.
(XLSX)

**S5 Data. Autoradiography signal distribution in CM and DMSs in MERS-CoV-infected cells.** CM, convoluted membranes; DMS, double-membrane spherule; MERS-CoV, Middle East respiratory syndrome-coronavirus.
(XLSX)

**S6 Data. Analysis of the autoradiography signal distribution around DMVs and DMSs in IBV-infected cells.** DMS, double-membrane spherule; DMV, double-membrane vesicle; IBV, infectious bronchitis virus.
(XLSX)

## Acknowledgments

We are grateful to Yvonne van der Meer, Charlotte Melia, and Barbara van der Hoeven for their assistance.

## Author Contributions

**Conceptualization:** Eric J. Snijder, Montserrat Bárcena.

**Data curation:** Montserrat Bárcena.

**Formal analysis:** Frank F. G. A. Faas, Montserrat Bárcena.

**Funding acquisition:** Eric J. Snijder, Abraham J. Koster, Montserrat Bárcena.

**Investigation:** Eric J. Snijder, Ronald W. A. L. Limpens, Adriaan H. de Wilde, Anja W. M. de Jong, Jessika C. Zevenhoven-Dobbe, Montserrat Bárcena.

**Methodology:** Ronald W. A. L. Limpens, Frank F. G. A. Faas, Abraham J. Koster, Montserrat Bárcena.

**Project administration:** Montserrat Bárcena.

**Resources:** Eric J. Snijder, Helena J. Maier, Abraham J. Koster.

**Software:** Frank F. G. A. Faas.

**Supervision:** Eric J. Snijder, Jessika C. Zevenhoven-Dobbe, Helena J. Maier, Abraham J. Koster, Montserrat Bárcena.

**Validation:** Montserrat Bárcena.

**Visualization:** Ronald W. A. L. Limpens, Montserrat Bárcena.

**Writing – original draft:** Eric J. Snijder, Montserrat Bárcena.

**Writing – review & editing:** Eric J. Snijder, Ronald W. A. L. Limpens, Adriaan H. de Wilde, Anja W. M. de Jong, Helena J. Maier, Abraham J. Koster, Montserrat Bárcena.

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
