## [Editor Report · Decision Letter 0]

10 Mar 2020

Dear Dr Bárcena, 

Thank you for submitting your manuscript entitled "A unifying structural and functional model of the coronavirus replication organelle: tracking down RNA synthesis" for consideration as a Research Article by PLOS Biology.

Your manuscript has now been evaluated by the PLOS Biology editorial staff, as well as by an academic editor with relevant expertise, and I'm writing to let you know that we would like to send your submission out for external peer review. I should warn you that we're not wholly persuaded of the strength of advance, so we'll be looking for some enthusiasm from the reviewers.

Please re-submit your manuscript within two working days, i.e. by Mar 12 2020 11:59PM.

Kind regards,

Roli Roberts

Senior Editor

PLOS Biology

---

## [Decision Letter · Decision Letter 1]

9 Apr 2020

Dear Dr Bárcena,

Thank you very much for submitting your manuscript "A unifying structural and functional model of the coronavirus replication organelle: tracking down RNA synthesis" for consideration as a Research Article by PLOS Biology. As with all papers reviewed by the journal, yours was evaluated by the PLOS Biology editors as well as by an Academic Editor with relevant expertise and in this case by three independent reviewers. The reviewers appreciated the attention to an important topic. 

Based on the reviews, we will probably accept this manuscript for publication, assuming that you will modify the manuscript to address the remaining points raised by the reviewers. Please also make sure to address the Data Policy and other policy-related requests noted at the end of this email.

We expect to receive your revised manuscript within two weeks. Your revisions should address the specific points made by each reviewer. In addition to the remaining revisions and before we will be able to formally accept your manuscript and consider it "in press", we also need to ensure that your article conforms to our guidelines. A member of our team will be in touch shortly with a set of requests. As we can't proceed until these requirements are met, your swift response will help prevent delays to publication.

*Copyediting*

*Published Peer Review History*

*Early Version*

*Submitting Your Revision*

Sincerely,

Roli Roberts

Senior Editor

PLOS Biology

DATA POLICY:

Regardless of the method selected, please ensure that you provide the individual numerical values that underlie the summary data displayed in the following figure panels as they are needed for readers to assess your analysis and to reproduce it: Figs 1D, 3ACD, 4CD, 5BDE, and S3CDE. NOTE: the numerical data provided should include all replicates AND the way in which the plotted mean and errors were derived (it should not present only the mean/average values).

REVIEWERS' COMMENTS:

Reviewer #1:

This manuscript by Snijder and co-workers entitled "A unifying structural and functional model of the coronavirus replication organelle: tracking down RNA synthesis" describes elegant studies aimed at discerning if double membrane vesicles or other ER sites modified by viral proteins are the site of coronavirus RNA synthesis during viral replication. The authors describe that an early study using immuno-EM revealed that virus-induced double membrane vesicles (DMVs) were the sites of viral RNA synthesis (Gosert et al., 2002 Replication of mouse hepatitis virus takes place at double membrane vesicles, reference 14). With the advancements in EM techniques and particularly EM tomography, researchers appreciated that coronavirus replication induced reorganization of host membranes, generating both convoluted membranes (CM) and DMVs (Knoops et al., SARS-coronavirus replication is supported by a reticulovesicular network of modified ER, reference 15). In addition, the term double membrane spherules (DMS) was introduced to describe membrane rearrangements detected during the replication of infectious bronchitis virus (Maier et al., 2013 IBV generates spherules from zippered ER membranes, reference 29), but it was not clear if these structures were unique to avian coronaviruses, or if they played a role in RNA synthesis. In this study, the authors used radiolabeling of viral RNA and detected the labeled RNA in association with the DMVs. Importantly, they also determined that CMs, DMSs, and DMVs can be visualized for all the coronavirus specimens they studied, including MERS-CoV, suggesting that these structures are generated during the assembly of the viral replication complex and the DMVs necessary for viral RNA synthesis. 

The technical work here is tremendous and the images of the viral protein-modified membranes are stunning. However, the manuscript seems have an inordinate focus on the DMSs, potentially implicating these as unique (terminal?) entities. Isn't is possible that the DMS detected at the moment of fixation is in fact on its way to becoming a DMV? Alternatively, some of these DMSs could be unable to progress to DMV status, for reasons that are not yet clear. In the absence of live-cell EM, it is not possible to distinguish these possibilities, but certainly the possibility the DMS to DMV transition could be considered as the authors develop their "unifying structural and functional model" of coronavirus RNA synthesis. Currently, there is a significant emphasis on the DMS in the manuscript, although this structure is not the take home message, and there is no functional role given to these structures, even as potential intermediates. If there is evidence that a DMS can not continue to grow into a DMV, then this should be made clear in the manuscript. Overall, the authors provide important results using state-of-the-art techniques that confirms and extends the early studies that that DMVs are the site of coronavirus RNA synthesis. 

Comments for the authors' consideration:

1. Lines 375-378: Consider changing the order of this description to 1) CM and zippered membranes; ii) DMS and iii) DMVs. This would allow you to speculate in a unifying model that coronaviruses modify membranes to promote "zippering", resulting in the formation of spherules that may mature into double membranes vesicles, the site of viral RNA synthesis. This "progression" is still hypothetical because it has not be visualized in real time, but certainly all the data presented here are at least consistent with this unifying model. One of the challenges is that researchers tend to see all of these structures simultaneously, and not a neat progression from one form to another. The discussion lines 410-413 seems to discount a progression model because of a lack of intermediate, but why couldn't a small double membrane spherule progress into a DMV? One could even speculate that the open DMS allows for capturing of a RNA that "seeds" the DMV to become a site of RNA synthesis. It is reasonable to speculate about potential models in the discussion section, and to be open to multiple interpretations of the current findings.

2. Line 381: change "establish" to confirms and extends. This manuscript works very hard at presenting uncertainty or controversy about coronavirus replication sites. However, the questions that arose about the sites of RNA synthesis came about as the techniques to visualize the modified membranes were improved over time, suggesting the potential for more than the originally documented DMVs were involved RNA synthesis. Perhaps this is a stylistic point, but pointing out that the results here confirm and extend earlier studies is important and accurate. 

3. The text could be much more concise by focusing on the idea that this work is evaluating all structures in multiple coronaviruses, which has not been done before in one manuscript, and seeking to determine if one or multiple sites are used for viral RNA synthesis. The "controversy" seems vastly over emphasized here. 

4. The images in the supplementary section are striking also. If PLoS Biology allows it, try to include all figures in the body of the paper and have only the movie in the supplement.

Reviewer #2: 

Coronaviruses have emerged as important zoonotic viruses with major threat to human health. In this paper, the authors characterized the ultra structures of human cells infected by different coronaviruses including MERS, SARS-CoV1, MHV, HCoV-229E and IBV. They have identified three different membranous structures, DMVs, CM and DMS, which have all been observed before in various coronavirus infections. The detection of DMS is new for MERS infection. Importantly, they were able now to provide a more universal picture that all these viruses induce the formation of all these RO structures. By using metabolic labeling of viral RNAs, after shutting down cellular transcription, they were able to demonstrate that DMVs are the likely places for coronavirus RNA synthesis and replication. Somewhat surprisingly, they find no significant roles for CM and DMS in coronavirus RNA synthesis and replication. This conclusion was further supported by detection of viral dsRNAs within DMVs. 

The findings are potentially important, especially the metabolic labeling data are quite exciting. I found the manuscript is fun to read with beautiful images. The manuscript requires the following changes. 

Fig. 1 shows images after 12 h incubation with many DMSs. What time point are these DMSs first appear?

Fig. 1 and Fig. 6. Please replace the arrowheads with different colors to aid the readers. Gray and black arrowheads could be quite similar on different computer screens.

Fig. 1F: please define what are arrows pointing at? 

Fig. 1F: Are the outer DMVs and the ER membranes are continuous membranes or are there gaps between them. It seems that they are continuous based on the presented images here.

Fig. 4B: it would be useful to pinpoint DMVs, since it is very crowded RO is shown here.

Fig. 5. It would be useful to know what % of label was inside versus outside of DMVs? It looks more label is on the outside of DMVs (although very nearby to the DMVs). What is the possible explanation? 

Lane 155: based on the findings in this work, I doubt it is correct to call zippered ER and CM as ROs, since no viral RNAs were detected in those structures.

L407: CM and DMS biogenesis does look very abundant at 12 time point in Fig. 1A-B, contradicting that these structures form late during infection. What is the first time point when CM and DMS structures are detected? The time of the vents for the biogenesis of various RO structures is confusing in this work.

Please discuss the origin of DMVs, since the ER origin of CM and DMS is discussed nicely in the discussion part. 

I note that DMSs are possibly too small to harbor the large coronavirus RNAs, possibly dsRNA (if that is the template for RNA synthesis), sgRNAs and the replicase complex, while copying the RNA templates. Even the 6x times smaller tombusviruses and nodaviruses form almost that size spherules to support replication as the observed size of DMS in this work.

Is it possible that DMSs are high-density membrane-protein (both viral and host-derived) storage granules to reduce ER stress in the cells?

Do the authors have data on how much percentage of the label is incorporated into genomic versus subgenomic RNA species during metabolic labeling? That would be useful information.

Reviewer #3:

In this manuscript, the authors performed a detailed structural and functional characterization of the replication organelles (ROs) of coronaviruses. Coronavirus ROs are composed of predominantly double membrane vesicles (DMVs). Through comparative analysis, the authors demonstrate that convoluted membranes, zippered ER and double-membrane spherules are conserved across alpha-, beta- and gamma- CoVs and altogether form the viral ROs. Additionally, the authors employed a classical, yet elegant approach that combines ultrastructural analysis with RNA labeling using radiolabeled nucleosides to demonstrate that viral RNA synthesis is associated with double membrane vesicles. This finding provides best available evidence so far that DMVs are the site of viral RNA replication. 

This is a timely study that sheds light on the role of the different membranous structures observed during CoV replication, although these structures have been described in several earlier reports. The main point of the present study is the clarification of a long-standing debate by showing that DMVs most likely are the site of viral RNA replication. 

Major comment:

The results of this study are clear and the author´s claims are supported by their data. One of the aims was to clarify the apparent differences reported in the ultrastructural architecture of the coronaviral ROs. The authors achieved their goal by showing that the ROs of the different CoVs are essentially the same. One of the limitations is that the authors provide a single snapshot looking at a single time point for the different viruses, often in different cells and using different MOIs. Moreover, the authors do not provide a growth curve of the viruses in the different cell lines. This information would be useful to allow for a direct comparison between the different conditions. The authors could at least comment if the selected time points reflect the initial phase of the viral infection or a later phase.

Additionally, what is the order of appearance of the different structures forming the CoV ROs and is this order conserved among the different CoVs? This could support the idea of a general mechanism used by all CoVs and the regulation of the biogenesis of their ROs. If such information is already present in the literature the author should comment on this point in the discussion. 

Minor points:

For the autoradiography experiments time points, MOIs and cell types differ between MERS-CoV and SARS-CoV infections. Are the time points selected by the authors comparable in terms of progression of viral infection of the two viruses in the different experimental conditions? In other words, are the selected time points within the exponential phase of viral replication or at the plateau phase of viral replication? The authors should comment on this.

Line 413 - there is no reference to backup this claim. Since this study does not analyze how the RO architecture changes during infection, the claim that DMV formation precedes the appearance of CMs should come from a previous study. The authors should add the reference of the paper to which they are referring.

FigS2- the figure legend states that Huh7 cells were infected with MERS-CoV while the figure annotation says Vero cells. The authors should clarify.

Line 177 - truncated sentence

---

## [Editor Report · Decision Letter 2]

14 May 2020

Dear Dr Bárcena,

On behalf of my colleagues and the Academic Editor, Andrea Cimarelli, I am pleased to inform you that we will be delighted to publish your Research Article in PLOS Biology. 

Early Version

PRESS 

Kind regards,

Vita Usova,

Publishing Editor 

PLOS Biology

on behalf of

Roland Roberts,

Senior Editor

PLOS Biology